# Academic Acculturation of International Doctoral Students in the U.S.: A Qualitative Inquiry

**DOI:** 10.3390/ijerph192316089

**Published:** 2022-12-01

**Authors:** Seungyeon Park

**Affiliations:** Department of Health, Physical Education and Exercise Science, School of Education, Norfolk State University, Norfolk, VA 23504, USA; sypark@nsu.edu; Tel.: +1-757-823-8455

**Keywords:** international students, doctoral program, academic acculturation, kinesiology

## Abstract

The aim of this phenomenological inquiry is to explore the academic acculturation experiences of international kinesiology professionals during their doctoral programs in higher education institutions in the U.S. Purposive sampling technique was used which include six study participants. The data collected from a demographic questionnaire and a focused interview. Using acculturation theory as a conceptual framework, the subjects’ academic acculturation as former international PhD students were described. There are three subthemes: (1) graduate program and requirements, (2) the academic environment of graduate program, and (3) professional development for a career in kinesiology. Findings are discussed in light of academic acculturation with a focus on assimilation (proactive adaptation in a new environment), and integration (flexibly to embrace the life experiences, learning experiences, and current experiences of the students in a new context). Particularly, findings examine how international kinesiology professionals perceived their doctoral degree experiences and sustained academic acculturation, pertaining to research and professional development at their programs. Exploring the academic acculturation of international PhD students is crucial for diversity awareness in higher education in the U.S. Often described as a minority with a narrower status, these international students undergoing academic acculturation should be assisted with aligning to their contextual frames, in terms of degree level and its characteristics in their field of study.

## 1. Introduction

Higher education institutions in the U.S. have been diversified by an increased number of international students. International students make up approximately 5.5% of the total number of students in U.S. higher education institutions [1]. In graduate program institutions in the U.S., international students make up approximately 19% of total enrollment [2]. It is estimated that a large number (80% of total enrollment) of graduate students in the areas of science, technology, engineering, and mathematics (S.T.E.M) are international students [3]. Although a smaller proportion of international graduate students compared to S.T.E.M fields, the total enrolment of international doctoral students has increased in the areas of social science and education [4]. The background and contributions of international graduate students have made U.S. higher education institutions more diverse [5].

There is a body of literature on international students in U.S. higher education institutions. Previous studies describe a variety of difficulties faced by international undergraduate and graduate students during cross-cultural adjustment. For instance, empirical research indicates that international students at all degree levels generally have difficulties due to lack of language proficiency, social disconnection, and intercultural maladaptation [6,7]. These difficulties negatively impacted academic performance and adjustment in higher education institutions in the U.S. [8,9,10,11] The distinctive learning and teaching culture in the U.S. can hamper the academic performance of international students, particularly during early phases of study. These students need time to adjust to their programs and their host country [12,13]. The literature on international students and their adjustment to the U.S. is frequently aligned with a deficit perspective, emphasizing various challenges such as language deficiency, difficulties in intercultural adaptation, and limited social relations in their schools and in daily life [14].

Exploring the degree experience of international students and their acculturation process should be more contextual. For instance, there are fundamental differences between undergraduate and graduate programs. Undergraduate students develop basic knowledge and academic skills through general and major courses to navigate their career before graduation. Doctoral students must polish their specialization by completing advanced courses involving formal research practices. It is a priority for doctoral level students to narrow their research topics and develop a methodological approach. Doctoral students must develop the ability to conduct their research independently and with advisors and colleagues by learning a wide scope of research procedures, including ethical standards for research practice. In the same vein, literature on doctoral students reports general concerns such as research development and/or involvement, dissertation completion, and academic or non-academic relations with advisors, faculty and colleagues [15].

More informative if research would focus on a specific discipline to explore the academic experiences of international students in the U.S., instead of focusing on an overall population of international students, regardless of their major programs and degree level. There is a need for an extended understanding of international doctoral students and their process for professional career development in terms of academic acculturation. Diverse factors impact successful degree completion: Not only individual aspects, but disciplinary culture and institutional culture are critical factors that impact successful academic experiences and acculturation at graduate programs for international doctoral students [16,17,18,19,20,21].

The aim of this phenomenological inquiry is to explore how international kinesiology professionals perceive their academic acculturation and doctoral degree completion in higher education institutions in the U.S. To attend to this inquiry, two primary aspects are examined: professional development as an international doctoral level student, and academic acculturation and intercultural experiences in their programs.

### Theoretical Framework and Purpose

The theoretical framework of this study is rooted in acculturation theory [22]. Acculturation theory calls attention to adjustment of ethnically diverse groups, including international students, who experience new contexts in a host country. Depending on whether subjects choose to maintain their previous perspective and cultural customs, or accept adaptations in a new context, acculturation is differently experienced. Research using acculturation theory has focused on psychological and attitudinal adaptation of various ethnic populations. For instance, previous literature using this acculturation approach explored daily lives of immigrants and the experience of international students in a new cultural context, how they interacted with a heterogeneous group of people, and the sociocultural factors [23,24]. This approach allows us to conceptualize international students and their professional development and/or career preparation. This study focuses on international doctoral students and their academic acculturation, and how they assimilate or dissimilate their identity as professionals in their field during their degree completion period. 

The following four specific tenets of acculturation theory are applicable to the present study [22]: Assimilation—active adaptation to sociocultural norms and systems in a new context.Integration—how students embrace both cultures by maintaining their own values and attitudes and accepting psychological and cultural norms over their original culture.Segregation—any rejection of intercultural changes by favoring their previous customs.Marginalization—individuals who mislay both their own and their host cultural affiliation.

Exploring academic acculturation of international students could broaden theory in higher education. Asian PhD students are underrepresented in their programs, and rarely appear in research about teachers of color [25,26]. To focus on academic experiences of international kinesiology professionals who would have experienced invisibility could empower the approach for professional development and doctoral degree completion. 

While previous research heavily focused attention on social and cultural aspects of international students, this study focused on professional development as doctoral level students. The purpose of this study is to explore the academic acculturation experiences of international kinesiology professionals, specialized in adapted physical education (APE), during their doctoral program in public research-intensive universities in the U.S. 

There were two guiding research questions: (a) what were academic acculturation strategies for Asian international professionals for completion of their PhD in kinesiology? and (b) how do Asian international doctoral students assimilate or dissimilate previous academic experiences into a new context at U.S. higher education institutions?

## 2. Materials and Methods

### 2.1. Recruitment and Participants 

The study uses the logic of both snowball and criterion sampling, which are purposive sampling methods [27]. Purposive sampling is a non-random, intentional strategy of selecting participants based on qualities matching the research purpose [28]. All six participants were Asian international professionals who achieved their PhD degree in kinesiology at a public university in the U.S. There are various major programs in the disciplines of kinesiology, thus, this study targeted participants with a major or minor in APE programs within kinesiology. This focus gives a more concrete lens contextually. Length of stay in the U.S. for participants ranged from 7 to 14 years. English is a second language for all the study participants. Table 1 is a summary of demographic information of study participants.

### 2.2. Methods

The study is based on phenomenological qualitative research using multiple case design. Multiple case design is robust method to conduct qualitative research based on empirical evidence [29]. To replicate study findings, researchers compare findings across all case of study participants [30]. To be more specific, the findings of each case were reviewed to identify similarities and then used to pull out a comprehensive understanding of academic acculturation experiences of international kinesiology professionals during their years in doctoral programs at U.S. universities. In the study, replication logic is equivalent for all individual research participants (e.g., research interest development), assuming they were placed in different conditions [29]. Research interest development is perceived as replication logic, and the study identified:A literal replication: research development as the common finding across each study participant.A theoretical replication: different challenges and strategies between participants.

### 2.3. Data Collection

Prior to data collection, participants submitted consent forms. Next, a demographic questionnaire was collected from study participants before a focused interview. The study used a modified version of Doctoral Education and Career Preparation for demographic information [31,32]. It took approximately 20 min for each participant to complete. This modified questionnaire had two main components: Part I, the experience as a doctoral student and Part II, the description of the doctoral program and department. Part 1 had two main categories. The first five questions had 12 sub-questions to identify the participant’s major, program, and program-related requirements. The other six questions had 45 sub-questions about academic support from the program and relations with faculty including academic advisors. Part II had 87 sub-questions under seven categories. These sub-questions examined the physical and social structure of the programs, their personal understanding of the program, the atmosphere of the program, and available resources while attending the program. 

The focused interview had three parts: (a) challenges and strategies connected to their doctoral degree experiences, (b) the academic learning experience and adjustment, (c) perspectives as an international professional connected to their doctoral degree completion in the U.S. Following approval from the Institutional Review Board, focused interviews were conducted with each study participant using on online platform. Prior to one week ahead of interview meetings, interview questions were sent to participants so that they had time to conceptualize their perspectives. The interviews lasted from 90 to 120 min. Several participants participated in follow-up phone calls for further interview questions if required. 

### 2.4. Data Analysis

The data collected from a demographic questionnaire and a focused interview. For data analysis, there were several steps. Initially, all interviews were transcribed by listening to recorded files of interviews. Next, researchers compared similarities to find common themes across study participants. More specifically, focused interviews were coded and thematized leading to subthemes using thematic analysis approach. Thematic analysis was the basic approach for analyzing the common themes for interviewed study participants. A researcher analyzed the data by segmenting and categorizing it to characterize the most integral concepts. Thematic analysis is a practical method to find common features across a data set of the study participants. More specifically, a sub-category was pulled out to represent common characteristics on the perspective of doctoral program experiences for Asian international individuals in kinesiology, in APE programs in the U.S. higher education universities. As the final step, all subcategories were reviewed using transcripts and a study of the participants’ perspectives and experiences, and were compared relating to the topic of this study.

### 2.5. Validity and Reliability

Multiple strategies were used to establish validity and reliability: triangulation, member checking, and peer-debriefing [33]. Accuracy of data was ensured using video interviews and transcripts of these video interviews (triangulation). The researcher shared individual data and interpretations of this data with each participant, and reviewed the data where there was any discrepancy (member checking). The researchers performed reviews to verify viewpoints and data interpretations (debriefing). Additionally, there are extensive descriptions about background and context for each participant for transferability of the study. Clear reporting was performed for all research procedures and methodology to establish dependability.

## 3. Results

This phenomenological inquiry explored academic acculturation experiences of international kinesiology professionals ascribed to their doctoral program in U.S. universities. The findings indicated that study participants had several common features. To provide rich descriptions of the findings of this study, we divided the findings into three subthemes: (a) graduate programs and requirements, (b) the academic environment of graduate programs, and (c) professional development for a professional kinesiology career.

### 3.1. Graduate Program and Requirements

All six participants described the graduate school and academic environment in U.S. higher education institutions. Particularly, there were many descriptions of the intense level of the graduate program curriculum and requirements. Hae shared his view about academic atmosphere:


*I think academic atmosphere is somewhat different. I experienced very tough level coursework requirements which required high criteria. Also, there was much emphasis for research expertise and training from the first semester. At the first glance, that was very intense. I needed to take lots of courses including advanced level of coursework including PE-based coursework as well as general APE courses. These include many topics like K-12 PE, various research methods, motor learning, psychology, and seminar courses during coursework. APE related courses were there like inclusion and APE, assessment for population with severe disability, and advanced APE.*


Another participant, Hui, explained about the increased burden, due to co-existing programs.

*Kinesiology doctoral program emphasizes training for faculty job at institutions of higher education. My understanding is that APE/APA is one of programs often coexisting with many other major areas like PETE* [physical educator teacher education], *sport psychology, sport management. Also some institutions provide APA concentration with other exercise science cores like physiology. So I felt like there are different foci depending major within kinesiology program, so those majors are concurrent but seems like orienting different directions where you study. This means that I sometimes needed to cover many other areas which I do not have background, not just studied my major area.*

Several participants described the requirements for APE doctoral degree completion. From demographic questionnaires and interview responses of study participants, APE doctoral degree completion generally included several steps; coursework for the major, a minor and/or cognate area, and a series of research method courses followed by qualification exam (written and oral), and dissertation phase (literature review and research proposal, data collection and analysis, results discussion and a dissertation defense). All participants shared similar experiences about this time consuming, highly focused, critical time in their life.

Several study participants, Paran, Hae, and Tan described institutional factors of APE graduate school, such as program characteristics and curriculum that make up the doctoral student’s experiences through different foci and training. For participant Paran, this pointed to the need for scrutiny when choosing an institution before starting a graduate degree:


*Within the areas of APA or APE, there are different approaches. This can be narrowed down depending on the final goal. This is also closely related to faculty position either as APE professional or exercise scientist (e.g., PETE program and Exercise Science Program). Different research foci made accordingly. A certain thing is there is something unique feature of APE doctoral program depending on different universities. When I chose the program, I need to review overall curriculum more specifically and tried to get information by contacting several faculty members.*


In addition to institutional factors, participant Kyo described the heavy workload of study and other duties, and high expectation at the doctoral degree program: 


*To enter doctoral program, international students required to provide GRE, TOEFL scores and pass GPA criteria. This was just minimum qualifications. My doctoral program and its training were so intense. Coursework included lots of different cores. On top of that, our program included some requirements like beyond coursework. For instance, I needed to complete grant proposal through one of courses. I spent much time to write grant works by combing my APE knowledge and conceptualizing ideas. A series of research courses were required as well. Every semester and all the parts were so intense and not simple during degree completion. Also, I need to complete multiple roles at the same time, I needed to spent so many hours per a week for my TA or RA duty. Assigned hours are just minimum and calculated hours but actual time use for work duties were much more. If I looked back on, it was meaningful, because I experienced field and many different values and areas. I spent time for supervision undergraduate preservice PE teachers in local public schools for multiple semesters. I went APE classes in the community with undergraduate preservice teachers for required hours.*


Two participants, Tan and Kyo, explained how they accumulated their APE knowledge during their program. Both graduate school coursework and field experiences broadened their perspectives and in-depth understanding. Due to overwhelming workloads and multiple responsibilities, study participants shared that they hesitated to devote multiple hours as teaching assistants and research assistants for data collection in public schools or community settings, even though it was required. However, they said the learning they gained through this field work proved to be an advantage. 

As students, participants had multiple requirements. Examples include serving children with disabilities, understanding legal amendments to the U.S. Individualized Education Educational Act (IDEA), and knowing the procedures of individual education plan (IEP) creation and implementation. One participant, Kyo, shared that he developed basic knowledge through coursework in APE contexts at the graduate program. As a doctoral student in an APE program, Kyo said he tried to critically reflect on how to enhance IEPs for APE subjects in an inclusive environment, actual learning from children with disabilities, and goal-setting with empirical data from research. Beyond this research perspective, he said the preK-12 public school setting was meaningful experience for refinement of knowledge. He was able to attend IEP meetings with approval from parents and school staff for research purposes. Kyo descried this experience:


*It was continuous process to lead parent’s final approval, I mean if parents disagreed about some parts, there should be another meeting with edited version of IEP. Based on kind of evaluation data, it was formal but, there were sometimes intense debate even between school staffs and parents. It was meaningful for me to observe IEP meetings how all the members tried to support child with special need by setting up detailed and realistic goals with the fact and data base, discussing about general classroom environment like total number of students and support need for students with disability. It was much more than contents that I learned by coursework and research. If I knew about overall concepts regarding IEP without field watching what is going on, it would be just one part. But this kind of field experience were so helpful for me to reflect and broaden my perspective. It was like piecing together a puzzle between graduate based academic learning and field-based reality issues.*


Participant Hui had a similar view. He explained there was synergy effect with graduate school coursework and life experience. He said: 


*There are many resources that we could know about APE and preK-12 in the U.S. I mean many things are doable through graduate program coursework and studying like through research paper, very general. But closer look for APE teacher’s involvement in IEP meetings, how they work with related to the present level of performance, set up annual goals and objectives, and planning and implementation… This is beyond books and studying. Maybe that is why PETE doctoral program require or prefer future candidate who have public school experience with teacher licensure.*


Two participants, Paran and Mi also shared the importance of their engagement in field experience. By working with in-service teachers, he said, he could see the nuances of K-12 and APE instruction, such as observing the child with special needs in services such as speech or occupational therapy, and their attendance in general classrooms with peers without disabilities. Additionally, one participant, Hae, shared he learned more about the realities of the APE filed in the U.S., such as possession of APE credentials. He supervised undergraduate students in several different schools in the same city, noted that all the schools are somewhat different in terms of available school and human resources. Several participants indicated that cross-cultural learning while connecting to their APE major program was helpful when learning understand the U.S. educational system for children with special needs.

All study participants indicated their doctoral program, and its many requirements were demanding at first. However, looking back, they found their degree completion and qualifications fulfillments were invaluable asset to them. Particularly, they obtained a balanced view by studying at a graduate program and experiencing K-12 public schools. This point of view became a crucial asset as international doctoral students in the context of their field of study.

### 3.2. The Academic Environment of Graduate Program 

Literature indicates doctoral degree completion is a consequence of many factors. Student traits and institutional factors are interconnected and influence the success of doctoral degree completion [34,35,36,37,38,39]. 

All participants emphasized the importance of advisors for the whole period of the doctoral degree. Some participants, Hae, Mi, and Kyo, talked about the need for careful selection of committee members for their qualification exam and dissertation phase. The academic advisor’s support and guidance were a particularly important factor from the start, directly and indirectly. Participants described it as their priority to get clear expectations through active communication with their advisors about academic progress, research development, and major-related knowledge. Academic advisors also assisted with choosing courses each semester, and facilitated research involvement. Interviewees said their advisors provided feedback beyond their academic area and also emotional support. Previous participant experiences with advisor relationships in their home countries tended to be more hierarchical, compared to the U.S. Participants needed adjust their relationships with academic advisors and professors in the U.S., since they are horizontal. Tan explained his experience:


*I cannot generalize my own experiences, but what I am feeling still is that academic advisor in my country works together closely like preparing APA workshop and undergraduate supervision for field experience, but still students in graduate program they need to work independently and then report based on hierarchical order. In the U.S., research fit (interest?) with academic advisor was the most important and priority when choosing doctoral program. As you know, research fit and any research interests in the APE should be the main considerations when finding advisor and program. I needed to narrow down my research topic and methodology. I am interested in physical activity for children with developmental disability in school-based settings. My advisor was more knowledgeable and have been worked regarding adult population with developmental disability in community settings. My program mostly oriented overall APA area, not school aged children and APE. Program curriculum within the area of kinesiology as APA or APE focus and advisor’s research area and methodology must be addressed when choosing school.*


Similarly, another participant, Paran, shared that he faced a dilemma because his advisor’s research foci were fully APE based, as was his, but the research methods were new for him. Another participant, Mi, talked about the structure of her doctoral program. She said faculty and graduate students tended to relate to each other well depending on the specializations they had in common within the program. Faculty only tended to support for their advisees or students working in the same major area. 

Similarly, APE is only a minor area in many programs in the U.S. Seminar courses, for instance, are heavily focused on a PETE core, not APE. These dynamics must be adjusted for and identified in each program. For some students, this is not optimal, but it is what is available, and they are left with no choice. 

Finding a program and an advisor that both align with the best interest and research fit is first and foremost for doctoral students. There are diverse specializations within kinesiology programs, and even within specializations. A prospective PhD student must understand details such as program curriculum, criteria, and requirements. 

### 3.3. Professional Development for a Career in Kinesiology

Doctoral students are required to develop their knowledge and expertise throughout research method courses, involvement of projects, and dissertation completion. All study participants shared that they continuously make an effort to strengthen their research expertise with the foundation of methodology and in the concentrated area they focused on. One participant, Hui described the varied approaches he took when he conceptualized his research focus in APE:


*To conduct quantitative method to evaluate population with developmental disability and physical activity pattern, there were many considerations in terms of measurement and assessment. Depending on choosing research population as stakeholders including caregivers and teachers or population with developmental disability or both, every step of research from literature review, methodology and analysis need to be different.*


Hui said that this seemed like a general approach for research development; however, he explained there can be trial and error when starting a graduate program. Hui emphasized during his interview that to him, “research” means “search, search, and research”. Several participants shared that many times they questioned themselves about what is most important in terms of PA or PE for populations with disabilities, beyond the research. Tan explained that he thinks research can be oriented to positive or negative aspects of the population with disabilities and their physical activity variables, and can reflect underlying perspectives. 

Some said they needed more expertise in methodology during their doctoral degrees. Due to multiple commitments such as coursework and graduate associate responsibilities, concentrating on higher level methodology courses was not always feasible. One participant, Kyo, described academic cultural differences between their home country and the U.S.:


*In the U.S., I needed to continuously update and build up my knowledge about the specific part of research. Focus was a certain thing. Compared to this, in my country, I had various approaches and trials about the field not merely limited to the specific type of disability. Research areas also included overall physical activity and health issues, not merely limited school PE. I mean I covered a lot of different areas at the same time. And I did not have any kind of limitation for this.*


Hae shared his perspective about differences between his home country and his U.S. experiences in academia: 


*Academic culture and approach are certainly different. For instance, publication here in the U.S. has more rigorous structures. Because I planned my research for school aged children with disability. I need to consume several months to get the final IRB approval. My advisor served as PI for me because, as you know doctoral students cannot be PI. I prepared IRB protocols with support of my advisor. There are many parts like literature review, research hypothesis and methodology and description for procedures and ethical areas. After submitted the first draft for IRB protocol, I needed to review and then update several parts including assessment methods for physical activity patterns for my topic, population with developmental disability. This process was like back to back revival patterns and time consumed a lot than I expected. But in my country, overall research process was much simpler, it is changing as more structured these days like the U.S. system but still even there is no need to clarify about IRB like the U.S. yet. So it was possible to make more products in short period time, but I am not sure about overall quality and its rigorousness in terms of research.*


Several participants (Hae, Kyo, and Tan) described differences in academic culture. The courses where these differences were more apparent were discussion-based courses, and they frequently served as a leading role. The U.S. academic atmosphere allowed them to gain more overall understanding of their field. For example, one participant shared she needed to know about the historical background of APE, such as rules and regulations around APE, and about K-12 public school systems for school-aged children with disabilities. She emphasized that it is important to acquire knowledge and background about APE, as well learning about current APE issues in the U.S. Because of the differences between their current and previous academic cultures, participants entered into study in a new context. They were required to adjust to a new academic environment, and to build up new knowledge attributed to a different social system in the host country.

Several participants expressed that they felt that their doctoral degree process was complicated, with many steps and burdens. For instance, students needed to be equipped with multiple tools for their coursework and dissertation, such as understanding reference management software and statistics packages. Study participants felt the doctoral degree completion process in the U.S. was overwhelming. Tan said, he thinks it would be impossible to do a part-time PhD in the U.S., while in contrast, there were many part time PhD programs in his home country. He recognized this as merely a difference, not a matter of right and wrong.

Tan shared that it was physically impossible to attend a PhD program as a part-time effort. For instance, the doctoral program required a great deal of preparation for coursework, such as thousands of pages of reading assignments per week for one course. Another participant, Kyo, said he felt like he studied more during one semester in the U.S. than during several years studying in his home country. Additionally, some interviewees experienced different academic structures such as the quarter system, which took some time to adjust to. Two participants, Paran and Kyo, said they felt academic pressure, but also had good momentum and motivation for their research, studying to satisfy the high expectations of their academic advisors. In contrast to their home country experiences, it was important to have more regular meetings with advisors, on a weekly basis, to closely discuss and gather corrective feedback on their work. This kind of academic rigor was a common feature in the responses of all the participants. Participants also pointed out that their graduate programs and their requirements were clear, and they were fully dedicated to satisfying these high criteria and expectations.

## 4. Discussion

The study explored experiences of international Asian doctoral students in the field of kinesiology, with an APE major: how participants assimilated, integrated, and were segregated and/or marginalized in their graduate programs, relating to their major study and research development [40]. The study extended upon previous literature to interpret the findings in relation to acculturation theory, its theoretical framework. The findings were described based on each subtheme in the following section.

### 4.1. Graduate Program and Requirements 

Within the field of kinesiology, study participants described their APE major, curriculum, and identity. There were several common features of the APE major study participants experienced in their home country and the U.S. The APE graduate program generally coexisted with the kinesiology department, and APE doctoral students generally had few APE courses during their doctoral degree completion. On the other hand, some mentioned they needed many other core courses, depending on the requirements of each program. There can be substantial differences in the number of courses required for APE graduate study, depending on the institution. APE major students can have from two to nine APE courses in graduate programs. Depending on the graduate program, its location, and the amount of time required for practicum, requirements varied [41]. 

Previous literature describes the academic burden of international doctoral students regardless of major programs. The data was not limited to language-related difficulties, but covered various challenges around the academic burden, and limiting social aspects of the effort [5,9,12,42]. In this study, international doctoral students expressed they were overwhelmed with heavy workloads in their programs. Students were required to take on multiple roles at once, such as work in research development, and doctoral coursework completion, and teaching or research assistant roles as a graduate associate. 

Study participants said graduate programs in the U.S. required many subcomponents. There were several steps including coursework completion, general exams, and dissertation proposal and defense, and each step required full attention to pass successfully. One participant shared about the difficulty of coursework completion, which included periods with several major courses and methodology courses. Not only did coursework completion require more than hundred credit hours, each course included many requirements. Seminar courses required a certain amount of time per week be spent in public schools. All study participants had graduate associate work, which required them to spend their time on assigned work, outside course completion and research development, to achieve their doctoral degree. These multi-faceted difficulties were unavoidable challenges for study participants during their intercultural experience. 

Several participants spoke about field work. At a first, they questioned the need for field-related participation while they were in graduate program. However, they did find it was helpful to broaden their understanding about the APE field and the U.S. educational system. Study participants pointed to this experience as essential. Study participants sought to continue their career as faculty in a U.S. higher education institution. They took on an assimilation strategy to better understand the APE field and the U.S. K-12 system. Similarly, as study participants experienced a new educational system in the U.S., they tried to integrate what they already knew to maximize their expertise in the field of APE. 

### 4.2. The Academic Environment of Graduate Program 

Participants emphasized the importance of their academic advisor for doctoral degree completion. This is consistent with the previous literature, which indicated the importance of academic advisors [43,44,45]. There are several explanations for the crucial role of advisor for a doctoral student. The academic advisor is a significant channel for information about the department and the field [43]. Academic advisors generally have a critical role on research practice and development, and career plans for doctoral students [46,47]. Beyond sharing information, each advisor’s dedication, provision of high expectations, and timely feedback were vital for the doctoral students [48]. Study participants chose their classes and developed research goals under the guidance of academic advisors from the beginning of doctoral program. Several participants said they needed to assimilate their communication methods with their academic advisors in the U.S., because the academic atmosphere and systems they experienced in their home countries were somewhat different. Most participants said they were accustomed to a hierarchical order between professors in their home country, while they progressed their research skillset more on a parallel basis with professors and academic advisors in the U.S. [49]. They said they could strengthen their expertise in research through regular meetings over multiple years during their doctoral degree period. In the same sense, their research fit with their academic advisors were considered the most integral part of their dissertation completion. During their doctoral program, the advisor’s academic support has an important role in helping students finish a series of requirements such as candidacy exams, and dissertation proposal and defense. To synthesize the findings from responses of study participants, the advisor’s role was important for meeting the standards of the graduate program for international Asian students. Good advisors ensured they were more easily assimilated into the new academic culture of graduate programs in the U.S.

### 4.3. Professional Development for a Career in Kinesiology

All the participants talked about academic culture and the research atmosphere in the U.S. In the findings, study participants shared that they felt that there were several emphases when attending in graduate program: Narrowing research topic down to one specialty and /or expertise in one specific area in terms of research and methodology, and achieving high quality during each step to achieve a doctorate.

Publication procedures were rigorous and time consuming: There were several steps from institutional review board (IRB) submission and approval prior to data collection and analysis. Overall, study participants explained that they were required to assimilate into this new academic atmosphere.

Although research procedures and work directions were similar in their home country compared to U.S. academia, study participants felt a difference in intensity. Although study participants actively searched for ways to assimilate into the new academic context, they faced an invisible glass ceiling. As a doctoral student, they needed to demonstrate that they were qualified to meet expectations in the program. However, because of vast differences in the social systems and atmospheres in academia, participants said they frequently needed more time than their domestic peers to fully adjust, beyond general challenges for an international, such as language proficiency and cultural expectations. For instance, some said they needed to build a new professional network for APE research. They also needed to integrate their previous knowledge about school systems with new experiences in the U.S. school system (e.g., APE in the U.S., IEP meeting requirements, APE and the public school system, and APE licensure across states).

Conclusively, participants had a chance to broaden their perspective about the APE field by critically thinking and comparing similarities and differences between their home country systems and the new context in their host country. Study participants said spending time in the field, such as work in public, allowed them learn how the U.S. system is organized. This series of requirements from doctoral program were practical and beneficial for international doctoral students seeking to assimilate into a new system and integrate their previous and current knowledge. This would be essential to work as professionals in higher education institutions.

### 4.4. Limitations and Implications

The findings of this study contribute to the body of literature about international doctoral students and their experiences at higher education programs. There are several practical implications from the findings of this study. First, diversity awareness should be discussed more in depth in doctoral degree programs of the U.S. higher education institutions. Faculty, staff members and students could better understand regarding unique features of international graduate students who would have different cultural and academic experiences. Second, international doctoral student should have to fully aware about their doctoral degree program such as curriculum, program requirements and research interests of faculty members before they start their doctoral program for academic acculturation.

There were several limitations of this study. This study focused on the experiences of international doctoral students in the areas of kinesiology. All study participants had major or minor concentrations based on APE. This strategy was intended as a specific focus for this study. Therefore, other disciplines may provide different feedback about the academic acculturation experiences of international doctoral students. To pull out common features of international doctoral students and their acculturation at their graduate programs, the study sought to exclude the acculturation process in participants’ daily lives. The target population was doctoral international students, because there were differences between doctoral and undergraduate programs, such as the purpose of degree completion, the priorities of the students, and curriculum. Study participants were all of international Asian origin, each of whom completed their doctoral degrees in previous years. However, there was no specific criteria for selecting study participants regardless of age, gender, and the length of stay in the U.S. The study recruited participants currently working as faculty in U.S. higher education institutions or their home countries. There is a possibility that these study participants were more likely to take an assimilation and integration approach to their new context during their doctoral degree programs in the U.S. All participants said they were fully focused on their adjustment and academic success, and considered continuing their careers in the U.S. However, if study participants had originally intended to get their degrees and return to their home countries, the findings would have also included separation and marginalization approaches by study participants, in terms of the four recognized types of acculturation theory. Due to the current public health pandemic emergency, it was important to follow recommended health policies. Because of this, we conducted all of the interviews by video or online. However, it was a convenient method to overcome the limitations of time and space, since the participants resided in different areas of the world.

## 5. Conclusions

The study explores the perspectives of international kinesiology professionals during their doctoral degree program in the discipline of kinesiology in the U.S. This study focused on one area of this specific discipline to explore how international doctoral students experience the acculturation process in their graduate program and academia in the U.S. Using acculturation theory as a conceptual framework, the findings indicated study participants used primarily assimilation and integration approaches during their intercultural academic experiences to make fast adaptation into a new academic environment as an international professional. There were three subthemes emerged: (1) graduate program and requirements, (2) the academic environment of graduate programs, and (3) professional development for a kinesiology career. Additionally, the findings indicated that the foremost goals of all the study participants were being more expertise in their field while they needed to handle multiple difficulties. Compared to this study, much of the existing literature is focused on international students but does not consider the types of degrees and major programs of the students. Furthermore, research has been aligned with a deficit perspective, by exploring the challenges of international student generally have. This study sought to explore the doctoral degree experience of international professionals based primarily on research development and professional expertise. In the future study, it will be needed to focus on the specific area of the disciplines to explore graduate program and international students, not merely focusing on general challenges and strategies that international students are likely to face.

## Figures and Tables

**Table 1 ijerph-19-16089-t001:** Participants Demographics.

Interviewee ^1^	Gender	Year ^2^	Current Position	Specialization ^3^	Citizenship
Hui	Male	11	Associate professor	APA ^4^, statistics	Korean
Paran	Male	10	Assistant professor	APA, statistics	Korean
Hae	Male	14	Assistant professor	APA, rehabilitation	Korean
Tan	Male	8	Assistant professor	APA, PETE	Korean
Mi	Female	10	Associate professor	PETE, APA	Chinese
Kyo	Male	9	Associate professor	Statistics, APA	Chinese

^1^ Pseudonyms; ^2^ Total number of years in the U.S.; ^3^ Expertise area(s) in kinesiology, ^4^ adapted physical activity.

## Data Availability

Not applicable.

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
