# Peer review of "Academic Acculturation of International Doctoral Students in the U.S.: A Qualitative Inquiry"

_ijerph, 2022, doi:10.3390/ijerph192316089_

Round 1
Reviewer 1 Report
The idea of using acculturation theory with a specific focus on kinesiology is interesting and the article overall is well-written. The research brings a unique perspective to light about populations who are underrepresented in the literature.
One area that would improve the results for a future study would be to find more participants as 6 is a small number and there is only 1 female included in the data.
In line 167 IRB is misidentified as the Initial Review Board, it should be Institutional Review Board.
In the data analysis section, how did you analyze the themes? More details are needed here.
The addition of the quotes from narrators strengthens this paper. The analysis of these quotes is strong and enhances the results.
The weakest section of the paper is the conclusion. It is too simple.
Author Response
Response: Thank you very much for your time and feedback. I updated my manuscript based on you feedback. You feedback is very important for me! Appreciate it again!
- In the future research, I will include more participations. Also, I agree on your feedback that it will be appropriate to include the same number of female and male research participant to make a balance. For your information, it was difficult to recruit more female participants for this study. However, I will make sure to include similar number of female and male participants in the future research to make study being more robust.
- Based on your feedback; in line 167 – Initial Review Board changed as Institutional Review Board
- In the data analysis section, how did you analyze the themes? More details are needed here.
- Response: I made changed in the data analysis section by adding more explanations.
(original manuscript) Initially, all interviews were transcribed by listening to recorded files of interviews. Next, researchers compared similarities to find common themes across study participants. Thematic analysis was the basic approach for analyzing the common themes for interviewed study participants. A sub-category was pulled out to represent common characteristics on the perspective of doctoral program experiences for Asian international individuals in kinesiology, in APE programs in the U.S. higher education universities. As the final step, all subcategories were reviewed using transcripts and a study of the participants’ perspectives and experiences, and were compared relating to the topic of this study.
(updated manuscript) The data collected from a demographic questionnaire and a focused interview. For data analysis, there were several steps. Initially, all interviews were transcribed by listening to recorded files of interviews. Next, researchers compared similarities to find common themes across study participants. More specifically, focused interviews were coded and thematized leading to subthemes using thematic analysis approach. Thematic analysis was the basic approach for analyzing the common themes for interviewed study participants. A researcher analyzed the data by segmenting and categorizing it to characterize the most integral concepts. Thematic analysis is a practical method to find common features across a data set of the study participants. More specifically, a sub-category was pulled out to represent common characteristics on the perspective of doctoral program experiences for Asian international individuals in kinesiology, in APE programs in the U.S. higher education universities. As the final step, all subcategories were reviewed using transcripts and a study of the participants’ perspectives and experiences, and were compared relating to the topic of this study.
- The weakest section of the paper is the conclusion. It is too simple.
- Response: Based on your feedback, I added more sentences by including the most important points of this study in the conclusion. Please a little excuse for me about the conclusion section. Regarding the conclusion part, my original writing included implications, limitations, and conclusions together. However, to match the journal’s template guideline, I separated conclusion part and implication/limitations part in different section.
(The original manuscript) The study explores the perspectives of international kinesiology professionals during their doctoral degree program in the discipline of kinesiology in the U.S. This study focused on one area of this specific discipline to explore how international doctoral students experience the acculturation process in their graduate program and academia in the U.S. Using acculturation theory as a conceptual framework, the findings indicated study participants used primarily assimilation and integration approaches during their intercultural academic experiences. Much of the existing literature is focused on international students, but does not consider the types of degrees and major programs of the students. Furthermore, research has been aligned with a deficit perspective, by exploring the challenges international student generally have. This study sought to explore the doctoral degree experience of international professionals based primarily on research development and professional expertise.
(Updated manuscript) The study explores the perspectives of international kinesiology professionals during their doctoral degree program in the discipline of kinesiology in the U.S. This study focused on one area of this specific discipline to explore how international doctoral students experience the acculturation process in their graduate program and academia in the U.S. Using acculturation theory as a conceptual framework, the findings indicated study participants used primarily assimilation and integration approaches during their intercultural academic experiences to make fast adaptation into a new academic environment as an international professional. There were three subthemes emerged: (1) graduate program and requirements, (2) the academic environment of graduate programs, and (3) professional development for a kinesiology career. Also, the findings indicated that the foremost goals of all the study participants were being more expertise in their field while they needed to handle multiple difficulties. Compared to this study, much of the existing literature is focused on international students but does not consider the types of degrees and major programs of the students. Furthermore, research has been aligned with a deficit perspective, by exploring the challenges of international student generally have. This study sought to explore the doctoral degree experience of international professionals based primarily on research development and professional expertise. In the future study, it will be needed to focus on the specific area of the disciplines to explore graduate program and international students, not merely focusing on general challenges and strategies that international students are likely to face.

Reviewer 2 Report
· A concise and factual abstract is required. It should be normally between 150 and 250 words. The abstract should state briefly the purpose of the research, the principal results, and major conclusions.
· In depth analysis and a more critical discussion of literature view is recommended. It can be more enriched and to be elaborated. The researcher’s own analysis should be added to reflect his/her own contribution.
· More updated references should be added urgently
· The social and practical implications/recommendations should be well highlighted and to be more elaborated
· The methodology and findings/conclusion can be more elaborated
Author Response
Reviewer # 2 Comments and Suggestions for Authors
A concise and factual abstract is required. It should be normally between 150 and 250 words. The abstract should state briefly the purpose of the research, the principal results, and major conclusions.
In depth analysis and a more critical discussion of literature view is recommended. It can be more enriched and to be elaborated. The researcher’s own analysis should be added to reflect his/her own contribution.
More updated references should be added urgently. The social and practical implications/recommendations should be well highlighted and to be more elaborated. The methodology and findings/conclusion can be more elaborated
Response: Thank you so much for your time and feedback. I updated my manuscript based on you feedback. You feedback is very important! Appreciate it again!
- A concise and factual abstract is required. It should be normally between 150 and 250 words. The abstract should state briefly the purpose of the research, the principal results, and major conclusions.
- Abstract was reviewed and then updated based on your feedback. Particularly, I added several sentences and then reorganized the structure to make a more concise and factual abstract based on your comments.
The aim of this phenomenological inquiry is to explore the academic acculturation experiences of international kinesiology professionals during their doctoral programs in higher education institutions in the U.S. Purposive sampling technique was used which include six study participants. The study participants were limited to Asian international professionals who earned their doctoral degree in the areas of kinesiology at a public university in the U.S. The data collected from a demographic questionnaire and a focused interview. Using acculturation theory as a conceptual framework, the subjects’ academic acculturation as former international PhD students were described. There are three subthemes; (1) graduate program and requirements, (2) the academic environment of graduate programs, and (3) professional development for a kinesiology career. Findings are discussed in light of academic acculturation with a focus on assimilation (proactive adaptation in a new environment), and integration (flexibly to embrace the life experiences, learning experiences, and current experiences of the students in a new context). Particularly, findings examine how international kinesiology professionals perceived their doctoral degree experiences and sustained academic acculturation, pertaining to research and professional development at their programs. Exploring the academic acculturation of international PhD students is crucial for diversity awareness in higher education in the U.S. Often described as a minority with a narrower status, these international students undergoing academic acculturation should be assisted with aligning to their contextual frames, in terms of degree level and its characteristics in their field of study.
- In depth analysis and a more critical discussion of literature view is recommended. It can be more enriched and to be elaborated. The researcher’s own analysis should be added to reflect his/her own contribution. The researcher’s own analysis should be added to reflect his/her own contribution.
- In the discussion and data analysis section, I made some changed or added several sentences. In the implication and conclusion section, I reviewed and made sure to pulled out some important points of this study as the perspective of the researcher of this study. Thank you so much for your feedback.
The data collected from a demographic questionnaire and a focused interview. For data analysis, there were several steps. Initially, all interviews were transcribed by listening to recorded files of interviews. Next, researchers compared similarities to find common themes across study participants. More specifically, focused interviews were coded and thematized leading to subthemes using thematic analysis approach. Thematic analysis was the basic approach for analyzing the common themes for interviewed study participants. A researcher analyzed the data by segmenting and categorizing it to characterize the most integral concepts. Thematic analysis is a practical method to find common features across a data set of the study participants. More specifically, a sub-category was pulled out to represent common characteristics on the perspective of doctoral program experiences for Asian international individuals in kinesiology, in APE programs in the U.S. higher education universities. As the final step, all subcategories were reviewed using transcripts and a study of the participants’ perspectives and experiences, and were compared relating to the topic of this study.
- More updated references should be added urgently. The social and practical implications/recommendations should be well highlighted and to be more elaborated. The methodology and findings/conclusion can be more elaborated
Response: Thank you so much for your time and feedback again. Also I ask for your understanding.
I reviewed and made sure to include appropriate resources in the updated manuscript regarding references. For instance, I tried to put the most recent information for the updated manuscript and put the most references which published within ten years.
In the implication part, I added several sentences.
- The findings of this study contribute to the body of literature about international doctoral students and their experiences at higher education programs. There are several practical implications from the findings of this study. First, diversity awareness should be discussed more in depth in doctoral degree programs of the U.S higher education institutions. Faculty, staff members and students could better understand regarding unique features of international graduate students who would have different cultural and academic experiences. Second, international doctoral student should have to fully aware about their doctoral degree program such as curriculum, program requirements and research interests of faculty members before they start their doctoral program for academic acculturation.

Reviewer 3 Report
The topic is interested.
Abstract. As few sentences may be added on methods adopted in this study.
A sentence or two may be added on the contribution of the study.
Methods
Section 2.1 line four mentions "all five participants" however, the following table mentions six participants. this may be corrected.
The methods section mentions about mix methods (survey and interviews), however the initail part (including abstract) of the paper is mentioning about only the qualitative methods, this may be corrected.
Regarding the survey questions, more details are needed on how these questions were developed, either adopted from previous studies or developed specially for this study.
The reliability and the validity of resarch questions should also be discussed in the methods section.
Details on interview questions are also missing, weather these were structured or un-structured interviews. Also add on the types of the interview questions.
The data analysis section is focused only on the qualltative data, explanation on the survey data analysis is missing.
The results and discussion section do not discuss about the outcome of survey.
The study contributions are not highlighed appropriately
Author Response
Reviewer # 3 Comments and Suggestions for Authors
Abstract. As few sentences may be added on methods adopted in this study. A sentence or two may be added on the contribution of the study.
Methods: Section 2.1 line four mentions "all five participants" however, the following table mentions six participants. this may be corrected. The methods section mentions about mix methods (survey and interviews), however the initail part (including abstract) of the paper is mentioning about only the qualitative methods, this may be corrected. Regarding the survey questions, more details are needed on how these questions were developed, either adopted from previous studies or developed specially for this study. The reliability and the validity of resarch questions should also be discussed in the methods section. Details on interview questions are also missing, weather these were structured or un-structured interviews. Also add on the types of the interview questions. The data analysis section is focused only on the qualltative data, explanation on the survey data analysis is missing. The results and discussion section do not discuss about the outcome of survey. The study contributions are not highlighed appropriately
Response: Thank you so much for your time and feedback. I updated my manuscript based on you feedback. You feedback is very important! Appreciate it again!
- As few sentences may be added on methods adopted in this study. A sentence or two may be added on the contribution of the study.
- Abstract was reviewed and then updated based on your feedback. Particularly, I added several sentences
and then reorganized the structure to make a more concise and factual abstract based on your comments.
The aim of this phenomenological inquiry is to explore the academic acculturation experiences of international kinesiology professionals during their doctoral programs in higher education institutions in the U.S. Purposive sampling technique was used which include six study participants. The study participants were limited to Asian international professionals who earned their doctoral degree in the areas of kinesiology at a public university in the U.S. The data collected from a demographic questionnaire and a focused interview. Using acculturation theory as a conceptual framework, the subjects’ academic acculturation as former international PhD students were described. There are three subthemes; (1) graduate program and requirements, (2) the academic environment of graduate programs, and (3) professional development for a kinesiology career. Findings are discussed in light of academic acculturation with a focus on assimilation (proactive adaptation in a new environment), and integration (flexibly to embrace the life experiences, learning experiences, and current experiences of the students in a new context). Particularly, findings examine how international kinesiology professionals perceived their doctoral degree experiences and sustained academic acculturation, pertaining to research and professional development at their programs. Exploring the academic acculturation of international PhD students is crucial for diversity awareness in higher education in the U.S. Often described as a minority with a narrower status, these international students undergoing academic acculturation should be assisted with aligning to their contextual frames, in terms of degree level and its characteristics in their field of study.
- Methods: Section 2.1 line four mentions "all five participants" however, the following table mentions six participants. this may be corrected. The methods section mentions about mix methods (survey and interviews), however the initial part (including abstract) of the paper is mentioning about only the qualitative methods, this may be corrected. Regarding the survey questions, more details are needed on how these questions were developed, either adopted from previous studies or developed specially for this study.
- Based on your feedback, I corrected and made changes.
- Section 2.1 line – I changed the sentence as ‘all six participants’
- This study is based on qualitative inquiry. There were two methods for data collection (a demographic questionnaire <modified questionnaire from the previous studies> and focused interviews). In line of 154 (as you indicated, I mistakenly wrote a sentence as ‘I used a modified survey’. Thus, I corrected this sentence as ‘the modified questionnaire in the sentence’. Thank you for your feedback.
- Also, I put references [31-32] to inform that the modified questionnaire was adopted from the previous research.
- Golde, C. M., & Dore, T. M. (2004). The survey of doctoral education and career preparation: The importance of disciplinary contexts. Paths to the professoriate: Strategies for enriching the preparation of future faculty, 19-45.
- Sato, T. (2007). Asian international doctoral students' assimilation into adapted physical activity graduate programs while attending predominantly white institutions of higher education within the Big Ten Conference[Doctoral dissertation, The Ohio State University].
- The reliability and the validity of research questions should also be discussed in the methods section. Details on interview questions are also missing, weather these were structured or un-structured interviews. Also add on the types of the interview questions. The data analysis section is focused only on the qualitative data, explanation on the survey data analysis is missing. The results and discussion section do not discuss about the outcome of survey. The study contributions are not highlighted appropriately
- Thank you for your previous feedback. I only used qualitative method for this study (focused interview and modified demographic questionnaire). Based on your feedback, I reviewed and made changes regarding interview-related information: (please see 2.3. Data Collection part). Also, I tried to describe the reliability and the validity of research by summarizing four approaches of qualitative method in section 2.5.
Data Collection
Prior to data collection, participants submitted consent forms. Next, a demographic questionnaire was collected from study participants before a focused interview. The study used a modified version of Doctoral Education and Career Preparation for demographic information [31-32]. It took approximately 20 minutes for each participant to complete. This modified questionnaire had two main components: Part I, the experience as a doctoral student and Part II, the description of the doctoral program and department. Part 1 had two main categories. The first five questions had 12 sub-questions to identify the participant’s major, program, and program-related requirements. The other six questions had 45 sub-questions about academic support from the program and relations with faculty including academic advisors. Part II had 87 sub-questions under seven categories. These sub-questions examined the physical and social structure of the programs, their personal understanding of the program, the atmosphere of the program, and available resources while attending the program.
The focused interview had three parts: (a) challenges and strategies connected to their doctoral degree experiences, (b) the academic learning experience and adjustment, (c) perspectives as an international professional connected to their doctoral degree completion in the U.S. Following approval from the Institutional Review Board, focused interviews were conducted with each study participant using on online platform. Prior to one week ahead of interview meetings, interview questions were sent to participants so that they had time to conceptualize their perspectives. The interviews lasted from 90 to 120 min. Several participants participated in follow-up phone calls for further interview questions if required.
- Based on your feedback and other reviewer’s comments, I made changes and added sentences in data analysis section.
The data collected from a demographic questionnaire and a focused interview. For data analysis, there were several steps. Initially, all interviews were transcribed by listening to recorded files of interviews. Next, researchers compared similarities to find common themes across study participants. More specifically, focused interviews were coded and thematized leading to subthemes using thematic analysis approach. Thematic analysis was the basic approach for analyzing the common themes for interviewed study participants. A researcher analyzed the data by segmenting and categorizing it to characterize the most integral concepts. Thematic analysis is a practical method to find common features across a data set of the study participants. More specifically, a sub-category was pulled out to represent common characteristics on the perspective of doctoral program experiences for Asian international individuals in kinesiology, in APE programs in the U.S. higher education universities. As the final step, all subcategories were reviewed using transcripts and a study of the participants’ perspectives and experiences, and were compared relating to the topic of this study.
